# Sociodemographic Determinants for the Health-Related Quality of Life of Patients with Vascular Amputations as Determined with the Prosthesis Evaluation Questionnaire

**DOI:** 10.3390/ijerph17082691

**Published:** 2020-04-14

**Authors:** José Vicente Benavent, José María Tenías, Ana Pellin, Jorge Casaña Mohedo, Ana Cristina Cabellos-García, Vicente Gea-Caballero

**Affiliations:** 1Hospital Lluis Alcanyis, Rehabilitation, 46800 Xátiva, Spain; jobecer@gmail.com; 2Dirección General de Investigación, Innovación, Tecnología y Calidad, Conselleria de Sanitat Universal i Salut Pública, 46010 Valencia, Spain; tenias_jma@gva.es; 3Valencia International University, 46002 Valencia, Spain; Ana.pellin@campusviu.es; 4Universidad Europea, 46010 Valencia, Spain; 5Hospital Universitari i Politècnic La Fe, Grupo de Investigaciόn Enfermero en Arte y Ciencia del Cuidado GREIACC, Health Research Institute La Fe, 46026 Valencia, Spain; 6Nursing School La Fe, Adscript Center of Universitat de Valencia, Grupo de Investigaciόn en Arte y Ciencia del Cuidado GREIACC, Health Research Institute La Fe, 46026 Valencia, Spain; gea_vic@gva.es

**Keywords:** amputees, physical therapy speciality, nursing, quality of life, artificial limbs, surveys and questionnaires, lower extremity

## Abstract

Background: To identify the sociodemographic variables independently related to the different dimensions of the Prosthesis Evaluation Questionnaire (PEQ). Methods: An observational, cross-sectional study was conducted, with a sample of 61 Spanish vascular amputees (Valencia, Spain). Included in this study are the results of the PEQ and explanatory-sociodemographic variables, as well as a descriptive and analytic analysis. Results: Gender differences were observed in “appearance” and “perception of appearance” (significantly higher levels for men). Older patients tended to have worse scores in “utility”, “frustration”, “social burden” and “deambulation”. More favorable scores were obtained for those residing in rural areas in “social burden” and “deambulation”. Educational level had a positive correlation with scores. Conclusion: Gender, age, place of residence, and educational level could be considered determinants of the quality of health related to prosthesis adaptation in vascular amputees. Clinical relevance: Knowing the influential variables in the process of prosthetization will allow better adaptation and an improvement in the quality of life.

## 1. Introduction

The loss of a lower limb not only negatively influences the quality of life of the patients. It also affects their social status and the purchasing power of the individual and the community [1], in all dimensions of the amputee’s personal, family and social life. This type of pathology poses an increasing social and economic burden, due to an escalation in the risk factors of amputation, such as diabetes and the aging population. 

The latest studies show that worldwide there are more than 202 million people suffering from peripheral vascular disease [2] involving the amputation of the lower limb (between 30,000 and 50,000 people a year in the USA). In the UK, an average of 8.2/100,000 amputations were estimated in 2014, while in Spain it was 7.6/100,000 in the same year [3], placing Spain as the second country in the world with the highest number of amputations after the USA, proportionally [4]. 

The most frequent indication of amputation is lower limb ischemia after having tried several revascularizations; the most frequent comorbidities are: diabetes mellitus (DM) (80.6%), coronary ischemic pathology (66.2%) and hypertension arterial (HTA) (68.2%) [5]. As Rubio et al. and Miller et al. [5,6] concluded, the diabetic population has a risk of suffering from a lower limb amputation between 10 and 30 times higher than the population that does not suffer from it.

Amputation of the lower limb has a great impact on physical and psychological well-being, mobility (along with an increase in falls) [7], and on the life and social function of individuals [8]. Moreover, it produces emotional disturbances and an increased pain level. In addition, in the surgical act itself, the mortality rate reached 16.8% [9]. 

The amputated patient is also a person who suffers a loss of a part of his body, so he enters the phase of grief that requires a relatively long period of adaptation; this is when the patient needs more professional care, since adaptation is linked to the emergence of both functional, and physical and psychological problems [10]. 

Regarding mobility, amputation is considered the biggest health event that negatively affects the patient’s mobility. The ability to walk with a prosthesis is of great importance for this type of patient, thus avoiding the negative impact on a physical, psychological and social level, also contributing to decreasing comorbidities [11]. This inability to walk causes a negative impact on performing basic day to day activities, the perception of body image and reintegration into the social environment; factors that seriously threaten the quality of life of the individual. After any type of amputation, the physician will usually propose the idea of a prosthesis; however, to achieve a good functional outcome, the work of the professionals, including those providing psychological support, must be well-coordinated and managed appropriately. In addition, patients receiving a prosthesis must be carefully selected [12]. 

Satisfaction with the prosthesis plays an important role in mobility recovery, prevention of rejection and increasing compliance with medical treatment. Between 40%–60% of amputee patients are not satisfied with their prosthesis; in this way, satisfaction with the prosthesis becomes a clear indicator of the quality of care and quality of life, playing an important role in the evaluation of the results of health care and a decrease in expenses. The objective is to place the prosthesis to recover mobility and improve the patient’s quality of life.

To measure the impact of a lower limb prosthesis on the quality of life, different factors have to be taken into account, such as support and social cost, the level of satisfaction with the prosthesis, mobility, level of activity or the presence of “phantom limb” pain, as well as the ability to learn how to use the prosthesis autonomously [13]. 

The Prosthesis Evaluation Questionnaire (PEQ) was described by Legro et al. in 1998 [14]; It was created with the intention of measuring the impact of a prosthesis on the quality of life of patients with prostheses. The PEQ is a self-evaluating instrument consisting of nine validated scales [15]: ambulation, appearance, frustration, perceived response, stump health, social burden, noise, utility, and well-being. It has been validated in more than five languages [12,16,17,18,19]. Previous studies with the PEQ questionnaire indicate a significant relationship between the level of infra- or supracondilea amputation and quality of life [20].

There is relatively few evidences of QoL impact on amputation due to peripheral diseases [21] and on the possible determining factors of QOL in amputees [22], especially on sociodemographic factors. In addition, there is no evidence on the relationship between the social determinants of health and the quality of life of amputees patients with prosthesis; this could also justify the study of structural determinants, such as the axes of inequality related to health (age, gender, social class, educational level, and residence [23]).

That is why the goal of the present study is to identify the sociodemographic variables that are independently related to the different dimensions of the PEQ in a representative sample of Spanish amputees fitted with a prosthesis.

## 2. Materials and Methods 

### 2.1. Design

This is an observational, analytical, cross-sectional study of a representative sample of Spanish vascular amputees residing in the autonomous region of Valencia, Spain, made in 2015.

### 2.2. Population

The study subjects were selected by means of consecutive sampling carried out at the Rehabilitation Services of the Lluís Alcanyìs Hospital in Xàtiva and the Ontinyent General Hospital, both in the Valencia Region. Of the total number of amputees in the study period (114), 61 were due to vascular causes (53.5%). The rest (53) were amputated for other reasons (tumor and traumatic), not being the subject of our study. 

The selection criteria were as follows: adult patients (>18 years of age), vascular amputees fitted with a prosthesis, with no significant cognitive deficits (not being clinically diagnosed in the patient’s medical history) or severe visual impairment (if the patient manifested visual difficulty in reading the questionnaire). 

The final sample consisted of 61 patients, who then underwent a recruitment visit in which they gave their informed consent to participate. 

### 2.3. Data Collection

After recruiting the study population, this was followed by a home visit to collect data on the patient’s quality of life and sociodemographic details, clinical data related to the patient’s particular vascular pathology and the process of prosthesization. 

The PEQ questionnaire is a self-administered instrument that measures the quality of life of people with vascular amputations. It consists of 82 questions (grouped into 9 independent scales); a numerical scale (Likert type) from 0 to 10 is used for assessment purposes. The PEQ has been shown to have adequate psychometric properties (Cronbach’s alpha over 0.8), which makes it suitable for Spanish-speaking prosthesis candidates.

Resulting variables (outcomes and end-points): the main outcome variable was the quality of life related to the prosthesis. This was estimated with the PEQ in its validated Spanish version [17]. A score was calculated for each of the dimensions of the questionnaire. 

Explanatory or independent variables: as possible determinants or factors associated with quality of life, we analyzed different variables related to the following items: 

The pathology and underlying process or processes that led to the amputation and subsequent fitting of a prosthesis.

Sociocultural and demographic aspects: age, gender, educational level, work situation, knowledge and use of the regional language, residential area (urban/rural), and coincidence of place of residence and place of birth (as a proxy variable for adaptation to the environment). 

All data were collected through a self-administered questionnaire completed during the same session, in which the PEQ was administered. 

### 2.4. Analysis Strategy 

Descriptive analysis. The various study variables were summarized according to their type: quantitative variables (age, scale scores) were summarized with central tendency measures (expressed as an arithmetic mean or median, depending on the distribution of values) and dispersion (expressed as the standard deviation or interquartile range), whereas qualitative variables (sociodemographic and clinical variables) were given in absolute and relative frequencies and expressed as percentages. 

Inferential analysis. The relationship between the different scales and the dimensions of the PEQ in relation to the sociodemographic and clinical variables was analyzed with the aid of non-parametric tests (the Mann–Whitney U test for comparisons between two groups and the Kruskal–Wallis test for comparisons between more than two groups). 

The level of statistical significance was set at *p* < 0.05. All data were analyzed with the aid of Statistical Package for the Social Sciences (SPSS) version 21, Spanish.

### 2.5. Ethical Considerations

All the subjects gave their informed consent to participate in this study; the study was approved by the ethics and research committee of both hospitals (Ethical Committee of Health Department Xàtiva-Ontinyent, number 26022009). The privacy of all participants was preserved by anonymizing all subjects, in accordance with Spanish and European data protection regulations. The authors declare the absence of conflicts of interest.

## 3. Results

Sixty-one patients were recruited, 44 men (72.1%) and 17 women (27.9%), with a mean age of 71.1 years (SD: 7.7 years; range: 51–87 years). The entire population was studied, so the response rate was 100%. 

Of the total number of evaluated patients, two-thirds lived in rural areas, with urban areas considered to be towns with 30,000 inhabitants or more. Most of the subjects were retirees who had only completed primary education. Although over half of the patients did not reside in their place of birth, most claimed to know and speak the local Valencian language (Table 1). 

The predominant underlying pathology was diabetes mellitus (46 subjects or 78%, all of them insulinized), with the most common level of amputation being supracondylar (53 subjects or 91%).

Table 2 summarizes the nine scales of the PEQ; the lowest scores were for the ambulation and frustration scales, while the highest scores were for noise and stump health.

The PEQ questionnaire, and sociodemographic/sociocultural adaptation variables. 

Gender differences were observed in the scales of appearance and perception of appearance, with significantly higher levels for men than for women (Table 3).

Older patients tended to have worse scores than younger ones in the scales relating to utility, frustration, social burden, and ambulation (Table 4). 

The type of residence was associated with significant differences in social burden and ambulation, with more favorable scores for those residing in rural areas versus urban centers (Table 5).

Educational level was also related to the values of several scales, with a tendency for higher scores for subjects who had completed secondary school compared to those who had not (Table 6).

The remaining variables (work situation, coincidence of place of residence and place of birth, knowledge and use of the regional language) were not significantly associated with any of the dimensions of the PEQ.

We can see in Figure 1 the summary of the independent and dependent variables (subscales) that have shown statistical significance.

## 4. Discussion

The quality of life of patients fitted with prosthetics is most aptly measured with specific instruments and scales such as the PEQ, which is adapted to the patients’ concerns and assesses the most relevant factors in the success or failure of the rehabilitation process. 

In this study, we have verified how certain variables, such as educational level, demonstrate a significant correlation with several domains of the PEQ. In contrast, domain scores were found to be relatively similar regardless of age, gender and other variables involving the patient’s adaptation to his or her environment. 

Our results are comparable to those of other studies which used the same instrument and explored the same variables and determinants.

For example, type of residence and living with relatives were both considered as variables in three of the four PEQ validation studies we examined [12,18].

The study carried out in the US showed that about one in five participants lived alone. Our study was the only one to differentiate between residence in urban and rural areas, with almost two thirds of patients (67%) residing in rural zones. In this sense, the contribution of our study is remarkable in determining that people living in urban environments obtained more unfavorable scores, and therefore will require more individualized processes to favor their adaptation, and a higher quality of life after fitting a prosthesis. 

The employment status of the subjects was another factor included in all the studies, except the one carried out in the Netherlands [6,9,11,15]. In all cases, most of the subjects were unemployed, which is not surprising considering the respondents’ ages and their disability. 

Cultural level was assessed indirectly through educational level. In both our study and that of Legro et al. [14], there was a predominance of patients who had only completed primary education (more than half), with the number of patients having completed secondary school and university being clearly higher in the American cohort. These differences can be explained by taking into account the age differences of the subjects in the various studies. 

The Italian study only provided the duration of education in years, with a range from 5 to 14 years of formal education. In our study, patients with basic education levels would be at risk of worse adaptation to fitting, and consequently, of lower levels of quality of life and satisfaction [6,15]. 

Patients’ adaptation to the environment was not comparable between the various studies. In the American study of Arwert et al. [12], for instance, this variable was estimated by calculating the frequency of various ethnic groups, with 15% of the subjects being Hispanic and/or African-American. In our study, the coincidence of place of birth and residence, along with the knowledge and use of the regional Valencian language, were used as proxy variables for adaptation to the environment. Still, while two-thirds of the subjects did not reside in their place of birth, most were still considered “highly” adapted to the area because of their knowledge of the regional language. Neither the Dutch nor the Italian study collected variables related to the cultural adaptation of the patients [9,18].

This may be of interest for the focus of the care plan, something that we consider important, since it has a special influence on the quality of life: women in our study obtained worse scores in the appearance and appearance perception dimensions, which is in line with other studies on self-concept, which state that men are more capable of functioning in social situations and are more satisfied with themselves and with life in general [24]; probably a good psychological support as well as the professional advice for the improvement of the physical image are pertinent. In the same way, we raise it in terms of the dimension of frustration in both sexes. These psychological needs are well documented, both professionally by the health team [25] and through mutual help groups, since the appearance of psychiatric disorders are frequent, such as dysfunctional grief, anxiety and depression, among others [26,27], responsible for the deterioration of the quality of life. 

In relation to grief, we highlight the need shown by patients to favor their psychosocial adjustment [26] and not to enter into dysfunctional griefs; that adjustment is impaired because, in reality, amputation has been a choice of curative treatment, and the individual must overcome the paradox that a curative treatment causes a deficiency, dependency, or severe adjustment problems to become independent again. This fact is minimized in vascular amputees, since they are convinced that amputation will improve their quality of life, and recognize the negative consequences of non-amputation. 

The best indicators that grief develops favorably will be a low presence of symptoms of anxiety or depression, good coping with body image changes, adequate social functioning with good adaptation to the social environment, and especially, we highlight the adaptation to the prosthesis. The objective of the health team (both rehabilitation and care) should be to direct its efforts toward the normalization of this symptomatology, since it will facilitate the acceptance and use of the prosthesis, consequently improving their quality of life [28].

Unfortunately, we could not find any previously published analyses of the associations between the various determinants and the levels noted in the dimensions of the PEQ. We were thus unable to verify whether the relationship between educational level and a better adaptation to the prosthesis is observed across various environments and cultures. The study and interpretation of these social determinants is of great interest. Our study explores how some social determinants of health proposed by the WHO may affect these patients [23]. In that sense, we have been able to identify how some known variables, which generate inequalities in health, influence the quality of life. The dimensions that have been most influenced by the determinants is the social burden (level of education, age, residence), followed by the perception of appearance, utility, and wandering. We highlight the influence of the level of education on well-being, as it is a key factor in adapting to new living conditions. This confirms in our sample that they are variables that identify a social profile of a patient at risk. 

It is interesting to note that we have not been able to find studies that relate these determinations to amputations of different origin: traumatic, tumoral, infectious. This fact reaffirms the need to deepen the knowledge of these determinants, which could guide us in the identification of people at risk for maladaptation, identifying that their origin is also social, as our exploratory study seems to show.

We believe that our results can guide professional interventions of a multidisciplinary nature, that will allow us to improve the adaptation of each patient according to their profile. This makes it easier for professionals to detect certain profiles of vulnerable patients, at risk of deteriorating their quality of life as a result of a potential and poor adaptation to their prosthesis. Therefore, this requires an exquisite planning of the prosthetization process as well as a good design of the multidisciplinary care plan that optimizes its adaptation. This proposal can be found in some studies developed both in community settings and in residential homes [29,30]. 

Our study had several limitations. The most noteworthy was the difficulty accessing the target population and therefore the difficulty of obtaining information about the population, and subjects therefore had to be interviewed one-on-one in their own homes. The location of each participant was obtained from several sources, among which were the orthopedic suppliers. Despite this, we have studied the entire population of vascular amputee patients that were implanted in our environment. Hence, we trust in the strength of our results. 

Patients were selected based on the activity records of the Physical Medicine and Rehabilitation Services of the two target hospitals in Xàtiva and Ontinyent, with most of the sociodemographic data and the information related to the underlying pathology and the health care received by each patient being extracted from the medical records of the respective hospitals. We encountered significant difficulties in conducting the surveys, as some patients were practically illiterate or had serious difficulties in reading. Thus, in almost all cases, we had to administer the questionnaires orally. Some subjects also showed a certain amount of cognitive difficulty in understanding and responding to some of the items. 

In addition, in our study eight subjects had an infracondilea amputation. Since the background indicates that the perception of quality of life may be different between the two groups, we could not compare these groups due to the small size of one of them. These poor outcomes may be due to the lack of conflict and the follow-up of diabetic patients by primary care nurses.

Still, more studies are needed in larger and more representative samples to verify both the direction and the magnitude of the found associations. It would also be useful to expand the number and diversity of factors to be assessed, including both those related to the patients, their families, and their social environments, as well as to the underlying pathology and the type of healthcare system (type of care center, availability of rehabilitation centers, etc.). 

Ideally, a multicenter study should be undertaken to cover various contexts and assess the impact of health systems on the quality of life achieved by these patients.

## 5. Conclusions

The practical implications of our study lie in the possibility of identifying a patient profile with a higher probability of success (or failure) in the process of prosthetization. Consequently, knowledge of the variables influencing the prosthetization process will allow for better adaptation and improvement of the quality of life; from our results, it follows that the level of education (most likely together with a greater cultural adaptation to the environment) can lead to a more favorable result and a higher quality of life for these patients. 

## Figures and Tables

**Figure 1 ijerph-17-02691-f001:**
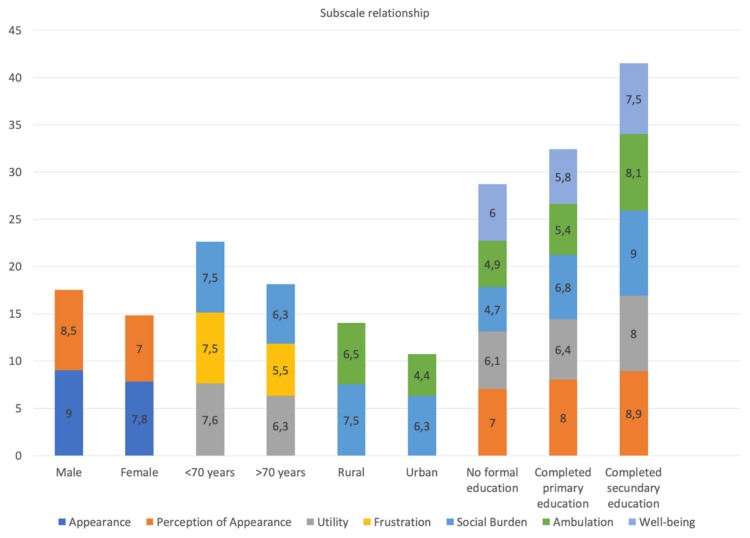
Relationship between statistically significant subscales and variables used.

**Table 1 ijerph-17-02691-t001:** Descriptors of the sociodemographic variables.

	*n* (%)
Residence Urban Rural	20 (32.8%) 41 (67.2%)
Cultural level No formal education Primary education Secondary education	13 (21.3%) 40 (65.6%) 8 (13.1%)
Work situation Employed Homemaker Retired	4 (6.6%) 12 (19.7%) 45 (73.8%)
Speaks regional language (Valencian) No Yes	16 (26.2%) 45 (73.8%)
Coincidence of residence with place of birth No Yes	34 (55.7%) 27 (44.3%)

**Table 2 ijerph-17-02691-t002:** Descriptive statistics of the various domains of the Prosthesis Evaluation Questionnaire (PEQ).

	Min	P25	Median	P75	Max
Utility	1.0	5.4	6.5	7.8	8.8
Appearance	5.4	7.6	8.6	9.1	10.0
Noise	4.5	6.8	10.0	10.0	10.0
Stump Health	4.3	8.2	9.2	9.7	10.0
Perception of Appearance	0.0	6.8	8.3	9.5	10.0
Well-being	0.0	4.6	6.3	7.5	10.0
Frustration	0.0	4.3	6.0	9.0	10.0
Social Burden	0.0	4.6	7.0	8.3	10.0
Ambulation	0.0	2.9	5.5	7.8	9.8

P25: percentile 25; P75: Percentile 75.

**Table 3 ijerph-17-02691-t003:** PEQ domain levels in relation to gender.

	Male (*n* = 44)			Femal (*n* = 17)			*p*
	P25	Median	P75	P 25	Median	P75	
Utility	5.2	6.6	7.8	5.8	6.5	7.6	0.87
Appearance	8.3	9.0	9.4	7.0	7.8	8.3	0.001
Noise	7.6	10.0	10.0	5.8	9.0	10.0	0.18
Stump Health	8.3	9.2	9.7	7.8	9.3	9.9	0.76
Perception of Appearance	7.0	8.5	10.0	6.0	7.0	8.5	0.046
Well-being	4.0	6.5	7.5	6.0	6.0	7.3	0.84
Frustration	4.5	6.3	9.6	2.3	5.5	8.5	0.35
Social Burden	4.7	7.0	8.9	4.6	7.0	7.8	0.82
Ambulation	2.9	5.5	8.1	2.9	5.5	7.3	0.85

P25: percentile 25; P75: Percentile 75.

**Table 4 ijerph-17-02691-t004:** PEQ domain levels in relation to patient age.

	<=70 Years (*n* = 26)			>70 Years (*n* = 35)			*p*
	P25	Median	P75	P 25	Median	P75	
Utility	5.8	7.6	8.0	5.1	6.3	7.3	0.02
Appearance	7.4	8.3	9.0	7.8	8.7	9.4	0.18
Noise	7.4	10.0	10.0	6.0	9.5	10.0	0.26
Stump Health	8.8	9.3	9.7	7.8	9.2	9.7	0.59
Perception of Appearance	6.2	8.5	10.0	7.0	8.0	9.0	0.70
Well-being	5.3	6.5	7.5	4.0	6.0	7.0	0.18
Frustration	5.4	7.5	10.0	3.5	5.5	6.5	0.04
Social Burden	6.2	7.5	9.0	4.5	6.3	7.5	0.01
Deambulation	4.3	7.4	8.2	2.1	5.3	6.2	0.01

P25: percentile 25; P75: Percentile 75.

**Table 5 ijerph-17-02691-t005:** PEQ domain levels in relation to area of residence.

	Rural (*n* = 20)			Urban (*n* = 41)			*p*
	P25	Median	P75	P 25	Median	P75	
Utility	6.2	6.9	7.5	4.3	6.4	7.8	0.24
Appearance	8.2	8.5	9.0	7.1	8.6	9.2	0.55
Noise	6.0	9.3	10.0	7.8	10.0	10.0	0.25
Stump Health	8.3	9.3	9.7	8.2	9.2	9.9	0.87
Perception of Appearance	7.0	8.0	8.5	6.4	8.5	10.0	0.49
Well-being	5.5	6.0	7.5	3.3	6.5	7.5	0.59
Frustration	5.5	6.0	9.5	1.5	6.0	8.0	0.18
Social Burden	6.6	7.5	9.0	3.3	6.3	7.5	0.02
Ambulation	5.1	6.5	8.1	1.9	4.4	7.0	0.02

P25: percentile 25; P75: Percentile 75.

**Table 6 ijerph-17-02691-t006:** PEQ domain levels in relation to educational level.

	No Formal Education (*n* = 13)			Completed Primary Education (*n* = 40)			Completed Secondary Education (*n* = 8)			*p*
	P25	Mean	P75	P25	Mean	P75	P25	Mean	P 75	
Utility	3.9	6.1	6.9	5.3	6.4	7.7	7.8	8.0	8.2	0.006
Appearance	7.2	8.4	9.4	7.3	8.5	9.2	8.5	8.8	9.0	0.89
Noise	6.8	9.5	10.0	6.5	10.0	10.0	7.3	10.0	10.0	0.86
Stump Health	7.6	9.5	9.8	8.2	9.2	9.8	8.9	9.3	9.6	0.89
Perception of Appearance	5.6	7.0	9.0	6.8	8.0	9.6	8.5	8.9	10.0	0.056
Well-being	3.3	6.0	7.3	4.6	6.0	7.0	7.1	7.5	8.3	0.059
Frustration	2.3	6.0	9.8	3.5	5.8	8.3	7.1	7.5	9.4	0.10
Social Burden	3.3	4.7	8.0	4.6	6.8	7.5	8.8	9.0	9.9	0.003
Ambulation	3.5	4.9	7.3	2.2	5.4	6.5	7.5	8.1	9.6	0.009

P25: percentile 25; P75: Percentile 75.

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
