# Peer review of "Sociodemographic Determinants for the Health-Related Quality of Life of Patients with Vascular Amputations as Determined with the Prosthesis Evaluation Questionnaire"

_ijerph, 2020, doi:10.3390/ijerph17082691_

Round 1
Reviewer 1 Report
The article tited: "Sociodemographic determinants for the health-related quality of life of patients with vascular amputees as determined with the Prosthesis Evaluation Questionnaire" is interestimg. Authors conclusions regarding their analisys are useful to identify a patient profile with a higher probability of success in the process of prosthetization. Also their results delivere the knowledge of the variables influencing the prosthetization process such as: level of education or cultural adaptation, which can lead to a more favorable result and a higher quality of life for these patients.
However, Authors did not join to their analisys medical factors such as ratio of amputation assessment. Corelation medical factors with patients feelling would be useful and more optimal. Authors are aware that ideally, a multicenter study should be undertaken to cover various contexts and assess the impact of health systems on the quality of life achieved by these patients.
Moreover, article is good wrriten, but I noticed one mistake in line 181. There is meaning of P75, which stands for percentile 7 instead percentile 75.
I accept thsis article to publish after minor revision.
Author Response
Please see the attachment.
We attach the article with the corrections suggested by the editors. We have marked in red the changes they have suggested, to facilitate their visualization.

Reviewer 2 Report
Thank you for allowing me to read your manuscript. I do have a few comments for the authors to consider:
Keywords
Please, all the keywords should be MeSH terms. Physioterapy, prosthesis or prosthesis evaluation questionnarie are not MeSH terms.
Material and Methods
Please specify further the characteristics of the measuring instrument (Prosthesis Evaluation Questionnaire). In the introduction section they explain that it consists of 9 scales, but not the response options as well as their interpretation.
Ethical considerations: please, include the reference number of the Ethics and Research Committee.
Author Response

(The authors gave the same response as above.)

Reviewer 3 Report
Introduction is too long, must be shortened
Exclusion criteria (no significant cognitive deficits or severe visual impairment): how were they evaluated?
The paper is disorganized.
Major concern regarding the limb and level of amputation. Due to this very important bias, I don't believe the paper is acceptable.
Author Response

(The authors gave the same response as above.)

Reviewer 4 Report
Introduction:
- The authors provided background information on why the effects of amputation on an individual's well being, and why it is a topic worthy studying in Spain. However, it'd like to encourage the authors to talk a bit more about previous studies on this topic. Are there previous studies on the social determinants of QoL and satisfaction among the study population of interest? What did they find? This discussion would provide some contexts for the readers. If there were no previous studies on this topic, state so.
Methods:
- A citation is needed for the statistical software used (SPSS v21).
- Is this study approved by an Internal Review Board (IRB) or an equivalent organization? If yes please state so.
Results:
- Line 160: Type II diabetes? Please clarify.
- The authors should provide more interpretations of the statistics presented in the tables. For example, in Table 3, what exactly did men and women differ in? Which sub-scale?
- The authors are recommended to use either dot or comma, not both, in numbers, to be consistent.
- The authors might consider presenting the mean sub-scales by the social determinant categories in bar charts or other graphs. This would help the readers better understand the how the QoL and sub-scales varied by the social determinants.
Discussion:
- Are there any reasons why the determinants of QoL among vascular amputees would be different from those for amputees of other reasons? Might be worthy discussing.
Minor issues:
- In the Abstract, the half sentence from in Line 25-26 is redundant.
- Better formatting and proofreading are needed. For example, one sentence as one paragraph is generally not encouraged in academic writing (Line 46).
- The Introduction and Results sections consist of a lot of very short paragraphs (1-2 sentences), which could be re-structured to form a better flow.
- Line 181, should be, P75: 75th percentile
Author Response

(The authors gave the same response as above.)

Round 2
Reviewer 4 Report
In Line 214, the author stated that the US study found 1 in 5 participants lived alone. What was the study sample of the US study? How was that relevant to this study? How does the living arrangement of people in the US inform your study? This part seems rather random.
Adjectives such as "remarkable" seems unfit for academic writing.
There are still a lot of 1-sentence or 2-sentence paragraphs. I suggest a scientific editor addresses this issue in writing style.